# Serotonin and Melatonin in Human Lower Gastrointestinal Tract

**DOI:** 10.3390/diagnostics13020204

**Published:** 2023-01-05

**Authors:** Rosa Vaccaro, Arianna Casini, Carola Severi, Antonietta Lamazza, Annamaria Pronio, Rossella Palma

**Affiliations:** 1Department of Anatomical, Histological, Forensic Medicine and Orthopedics Sciences, Sapienza University of Rome, 00161 Rome, Italy; 2Department of Translational and Precision Medicine, Sapienza University of Rome, 00161 Rome, Italy; 3Department of General, Plastic and Urologic Surgery, Sapienza University of Rome, 00161 Rome, Italy; 4Department of General Surgery and Surgical Specialties, Sapienza University of Rome, 00161 Rome, Italy

**Keywords:** enterochromaffin cells, serotonin, melatonin, lower gastrointestinal tract, inflammatory disease, ulcerative colitis

## Abstract

Background and Aims. Melatonin is a ubiquitous hormone produced not only by the pineal gland but also by other organs and tissues. It is involved in the regulation of several gastrointestinal functions. The main cells responsible for the production and release of extrapineal melatonin are the enterochromaffin (EC) cells that produce serotonin. They are involved in the pathogenesis of neuromotor disorders that characterize functional gastrointestinal disorders and in the pathophysiology of inflammatory intestinal diseases. Our aim was the immunohistochemical highlighting on biopsy samples of normal gastrointestinal mucosa and in ulcerative colitis (UC) of immunoreactive cells for melatonin and serotonin in order to identify any differences in their distribution. Materials and Methods. Our prospective case-control study involves the highlighting on human mucosal biopsies of immunoreactive cells for melatonin and serotonin. All patients undergoing colonoscopy + ileoscopy were considered eligible for the study, divided into two groups: 1. patients with active ulcerative colitis (UC); 2. control group consisting of patients undergoing endoscopic examination for colorectal cancer screening. Results. Twenty-one patients were enrolled. The controls had a higher concentration of EC cells containing 5HT particularly in the rectum (*p* value ≤ 0.05). In patients with active colitis the expression of 5-HT-iR was greater in all tracts of the colon. The correlation analysis in UC patients shows that a higher expression of 5-HT-iR+ cells corresponds to a lower extension of the disease and a greater severity of the same. Conclusions. 5HT+ cells decreased in the case of UC compared to healthy controls. In the severe disease, there was an increase in the expression of melatonin-secreting cells, probably as a compensatory response to the inflammation and oxidative stress. This increase is negatively correlated with the extent of the disease and positively with the severity of the same.

## 1. Introduction

Melatonin is a ubiquitous hormone produced not only by the pineal gland but also by other organs and tissues, especially in the gastrointestinal tract, where the concentration of melatonin has been estimated to be approximately 400 times greater than in the pineal gland [1]. Melatonin is a hormone involved in the regulation of several gastrointestinal functions [2,3] through its antioxidant activity as a scavenger of reactive products, modulating the activity of antioxidant enzymes and stimulating the immune system response [4], and also playing a role in tissue repair mechanisms in different clinical conditions [2].

Furthermore, it has been demonstrated that dysregulation of circadian rhythm is associated with motor alterations of the colonic mucosa and with mechanisms of cell destruction. These properties, related to the clinical evidence of low levels (plasma, urinary, salivary fluids) of melatonin in patients with functional and organic gastrointestinal disorders compared to healthy controls [5,6] support the involvement of melatonin in their pathogenesis. For this reason, it has been studied as an adjuvant treatment in several gastrointestinal diseases such as irritable bowel syndrome (IBS), chronic inflammatory bowel diseases (IBD), and necrotizing enterocolitis [7]. Our preliminary data [8] on both upper and lower GIT showed that EC cells storing 5HT (expressed as percentage of mucosal lining cells) were present from gastric antrum to the rectum, the highest concentration being found in the stomach (7.3 ± 3.37%). EC cell presence remained stable in the ileum (4.03 ± 0.91), decreased in the colon (2.57 ± 1.4), and increased again in the rectum (3.8 ± 1.8). Melatonin immunoreactivity in turn was never clearly evidenced in 5HT immunoreactive EC cells but lined the surface of the mucosa and lumen of intestinal glands. It then appears that, differently from 5HT, melatonin is not stored, but immediately released upon biosynthesis into the extracellular fluid and circulation. In order to understand the potential role of melatonin distribution in human GIT mucosa in inflammatory gastrointestinal diseases we drew our attention to patients affected by UC.

### 1.1. Biosynthesis

Melatonin is synthesized during the night by pinealocytes starting from serotonin by the action of two enzymes involved in its biosynthesis (AANAT and HIOMT). In the gastrointestinal tract, the main cells responsible for the production and release of extrapineal melatonin during the day are the enterochromaffin (EC) cells that produce serotonin [1,9]. EC cells are involved in the pathogenesis of neuromotor alterations that characterize gastrointestinal functional disorders and also appear to be involved in the pathophysiology of inflammatory bowel diseases [10,11].

Serotonin (5HT) is produced by gut enterochromaffin (EC) cells and is involved in the pathogenesis of neuromotor disorders that characterize functional and inflammatory gastrointestinal diseases. Melatonin is a synthetic product of both the vertebrate pineal gland and the EC cells of the gastrointestinal tract (GIT). Both melatonin and serotonin share the same biosynthetic pathway from tryptophan, but their effects are almost antagonistic. Melatonin in the GIT serves as a natural inhibitor of serotonin action on peristalsis, serotonin-induced spastic contractions of rat ileum being reverted by melatonin. In addition, melatonin is a powerful antioxidant that modulates the immunological GIT system. Studies on the distribution of EC cells in the normal human GIT date back to the nineteen-seventies and, although melatonin has been found in mammal and human gastrointestinal tracts, to date there are not clear morphological data on the distribution of melatonin immunoreactivity in normal human gastrointestinal mucosa and on its relationship with the serotoninergic system.

The quantitative study, on colonic and ileal mucosal biopsies, of the expression and distribution of melatonin and its receptors could have an important clinical impact in patients suffering from both functional and organic pathologies of the lower gastrointestinal tract.

### 1.2. The Role of Melatonin in Ulcerative Colitis

The incidence of UC is estimated to be 9–12 per 100,000, with a prevalence of 205–240 per 100,000 [12]. Moreover, approximately 20 percent of people with UC have a close relative with IBD [13]. UC is a chronic disease with periods of exacerbation and remission. The pathogenesis of UC is complex, involving inflammatory and immune factors, endothelial barrier dysfunction, and the overproduction of reactive oxygen forms, all of which play a role in the destruction of colonic mucosa [14].

It has been shown that Th2 and Th17 pathways are predominant in UC [15]. In addition, IBD-specific changes in the gut microbiota play an important role in UC disease [16,17,18,19].

In a previous study [20] the authors compared the expression of the melatonin-synthesizing enzymes tryptophan hydroxylase (TPH1), arylalkylamine-N-acetyltransferase (AANAT), and *N*-acetylserotonin methyltransferase (ASMT) in the colonic mucosa and urinary excretion of 6-sulfatoxymelatonin in patients with UC and lymphocytic colitis (LC). The expression of TPH1, AANAT, and ASMT in colonic mucosa in UC and LC patients was significantly higher than in healthy subjects. These results indicate that patients with UC and those with LC may display high levels of melatonin-synthesizing enzymes in their colonic mucosa, which could possibly be related to increased melatonin synthesis as an adaptive antioxidant activity.

## 2. Materials and Methods

Our prospective case-control study provides, in collaboration with the Human Anatomy Laboratory, the highlighting on human mucosal biopsies of enterochromaffin cells that produce melatonin in healthy patients and in patients with UC.

All patients who underwent rectosigmoid colonoscopy and retrograde ileoscopy for screening for UC were considered eligible for the study. The study was performed from 2019 to 2021 at the AOU Policlinico Umberto 1—Sapienza University of Rome. The study was conducted in accordance with the Declaration of Helsinki and the principles of Good Clinical Practice. Written consent was obtained from each subject enrolled in the study. Ethical approval for this study was obtained from AOU Policlinico Umberto 1—Sapienza University of Rome (approval number: 6244).

### 2.1. Patient Selection

All the outpatients were evaluated by the same gastroenterologist who assessed their clinical status and the indications for endoscopy. Pre-endoscopic questionnaires were administered in order to identify eligible patients who were divided into two study groups:Control group consisting of healthy patients undergoing routine endoscopic examination for colorectal cancer screening in absence of symptoms and/or warning signs;Patients with active ulcerative colitis (UC) in treatment or in remission according to the AGA 2019 guidelines [21].

Pre-endoscopic clinical evaluation in UC patients was performed using the Mayo Score for Ulcerative Colitis Activity [22].

The biopsy samples were collected in duplicate (histological examination and immunohistochemical procedure) according to the following scheme:

Ileo-colic mucosa:Ileus: at least two biopsies (1 for histology and 1 for immunohistochemistry);Right colon: at least four biopsies (2 for histology and 2 for immunohistochemistry);Transverse colon: at least four biopsies (2 for histology and 2 for immunohistochemistry);Descending colon: at least four biopsies (2 for histology and 2 for immunohistochemistry);Sigmoid colon: at least four biopsies (2 for histology and 2 for immunohistochemistry);Rectum: at least four biopsies (2 for histology and 2 for immunohistochemistry).

### 2.2. Immunohistochemical Analysis

The biopsy samples were fixed in a 4% solution of paraformaldehyde in 0.01 M phosphate buffer saline (PBS) for 24 h at +4 °C. They were then washed in 80 °C alcohol, dehydrated, and embedded in paraffin; 7 µm thick sections were made which were mounted on slides treated with glycerin albumin. In each slide, 4 sections were mounted at a distance of approximately 500 µm away from each other. The sections were then rehydrated to perform routine staining with control hematoxylin-eosin or to be processed by immunohistochemical method. To deactivate the endogenous peroxidase, the sections were previously treated with a solution of PBS with the addition of 0.1% sodium azide and 0.5% H_2_O_2_ for 30 min at room temperature. To avoid nonspecific antibody binding, the sections were pre-incubated with normal goat serum (NGS Vector Laboratories, Burlingame, CA, USA) diluted 1:30 in PBS containing 1% bovine serum albumin (BSA; Sigma, Saint Louis, MO, USA) for 30 min in a humid chamber, at room temperature. Subsequently, the sections were incubated with polyclonal anti-serotonin antiserum (Chemicon International, Temecula, CA, USA), diluted 1:10,000, in a humid chamber for 48 h at +4 °C. After washing in PBS, the sections were incubated with biotinylated anti-rabbit sheep antiserum (Goat anti Rabbit, Vector Laboratories, Burlingame, CA, USA), diluted 1:1000 in PBS, in a humid chamber, for 1 h at room temperature. After washing in buffer, the sections were incubated with a streptavidin-biotin-peroxidase complex (ABC, Elite Kit, Vector), diluted 1:2000 in 0.05 M Tris-HCl buffer, pH 7.6, in a humid chamber for 1 h at room temperature. After washing in Tris-HCl, peroxidase activity was evidenced by reaction with a solution containing 0.04% 3-3′diaminobenzidine tetrahydrochloride (DAB; Fluka, Buchs, Switzerland), 0.4% nickel-ammonium sulfate, and 0.003% H_2_O_2_ in 0.05 M Tris-HCl buffer, pH 7.6, for 3 min at room temperature. For specificity control the primary antiserum was substituted with PBS alone or with suitably diluted normal rabbit serum. The sections were dehydrated and mounted with coverslips for microscope observation.

### 2.3. Cell Count

Four fields were acquired for each section and a count of total enterocytes and serotonin immunoreactive endocrine cells was performed. The ratio of 5HT-ir endocrine cells/total enterocytes was obtained for each field. In the terminal ileum, acquisitions of separate fields and counts were made for the villi and glands. An average was made of the values obtained in the fields of the same section and in the different sections of the same sampling.

### 2.4. Statistical Analysis

All collected data in this study were reported using summary tables and/or data lists. A correlation analysis of the Pearson coefficients (rho) was performed. Binary variables representing the various values considered were created using the one-hot encoding technique. All tables, their values, and p-values were obtained using the Python pandas, matplotlib, scipy, and giotto-time libraries. The correlation coefficients were considered statistically significant for *p*-values ≤ 0.05.

## 3. Results

Twenty-one patients were enrolled (12 in the control group, 9 in the UC group), mean age: 53.8 years, female: 6. Demographics and endoscopic characteristics are summarized in Table 1.

In the UC group four patients were in remission (Mayo 0–1) and five patients presented active disease (Mayo 2). Both serotonin and melatonin immunoreactivities were found throughout ileocolic mucosa with different patterns of distribution.

The current data show that the controls present the highest concentration of EC cells storing 5HT. It appears physiologically more expressed in the rectum (5.8 ± 3.43%) and in the ileal glands (3.52 ± 0.99%). This concentration is significantly higher in the rectum (*p* value ≤ 0.05) (Figure 1). 

However, in both control and UC groups, 5-HT immunoreactive EC cells are present from the ileum to the rectum, the highest concentration being found in the rectum (5.8 ± 3.43%) for controls and in the ileal glands (3.59 ± 1.63%) for UC (Figure 2).

We then compared patients with UC in remission (Mayo 0–1) with patients with active UC (Mayo 2) and 5-HT-Ir was found to be different. In particular, we found that in patients with active colitis (Mayo 2) the expression of 5-HT-iR was greater in all parts of the colon, particularly in the sigmoid colon, although this difference was not statistically significant (*p*-value = 0.06) (Figure 3).

### Correlation Analysis

The correlation between the concentration of 5-HT-iR and the extension of the disease, according to the Montreal classification (E1,E2,E3), was estimated by determination of Pearson’s correlation coefficient (rho), a linear regression equation. Pearson coefficients measure correlation of pair of variables and assume values between −1 and 1. In our context, a positive correlation with the extension of the disease can be interpreted as a higher expression of 5-HT-iR and a negative correlation as a reduction. Correlation coefficients were considered statistically significant for *p*-values less than or equal to 0.05.

Binary variables representative of the various values considered were created using the one-hot encoding technique. All the tables, the corresponding values and *p*-values were obtained using the Python libraries pandas, matplotlib, scipy, and giotto-time.

A negative correlation was found between 5-HT-iR and the extension of the disease. This negative correlation was found in all the investigated ileal and colonic tracts but particularly for the localization “ileum” (rho = −0.756) and “descending colon” (rho = 0.758). More concretely, a high positive Pearson coefficient indicates a variable for which a higher value corresponds to a higher extension of the disease. Conversely, a high negative Pearson coefficient is found for a variable for which a higher value corresponds to a lower extension of the disease. In our data in the patients affected by UC, a higher concentration of 5-HT-iR corresponds to a lower extension of the disease, while a lower concentration of 5-HT-iR corresponds to a higher extension of the disease (Figure 4).

We then performed a correlation analysis between the severity of the disease and the expression of 5-HT-iR cells. In this case, on the other hand, a positive correlation was found in all the analyzed tracts and in particular in the sigma (rho = 0.93). This correlation indicates that an increase in 5-HT-iR cells positively correlates with disease activity in UC patients (Figure 5).

## 4. Discussion

The results obtained confirm our previous observations on both upper and lower GIT which showed that EC cells containing 5HT (expressed as a percentage of mucosal lining cells) were present from the gastric antrum to the rectum [8]. The anti-inflammatory role of GI melatonin is well known. The melatonin can modulate the immune response by inhibiting macrophage activity through the reduction of NF-κB levels, COX-2 and iNOS activity; furthermore, it modulates prostaglandin E2 secretion and regulates gene expression of proinflammatory cytokine levels including interleukins (IL-1), tumor necrosis factor alpha (TNF-α), and IFN-γ [23,24,25,26]. Gastrointestinal melatonin has antioxidant effects, reduces the degradation of prostaglandins by prostaglandin reductase, and limits gastric lesions and hydrochloric acid secretion; it also antagonizes the actions of 5-HT, which are linked to the formation of gastric ulcers [27,28,29].

Considering the well-known role of melatonin in tissue inflammation processes and thus focusing on UC patients, we can conclude that the inflammatory processes in the colon are accompanied by an altered secretion of melatonin. A decrease in plasma levels of melatonin and urinary excretion of aMT6 has been observed in previous studies in patients with ulcerative colitis [5]. A particularly important result of our study is that the concentration of immunoreactive cells for serotonin is significantly lower in UC patients than in healthy rectal controls.

On the other hand, when patients are stratified according to disease activity, 5HT-iR values increase in patients with active disease (Mayo 2) compared to healthy controls, although they do not reach statistical significance. It could be explained as a compensatory increase in the expression of melatonin in response to the inflammatory process. However, these beneficial changes in melatonin homeostasis are not sufficient to inhibit the inflammatory process and to achieve spontaneous remission of the disease. In fact, remission requires the administration of many drugs, including anti-inflammatory and immunosuppressive ones, because the pathogenesis of ulcerative colitis is complex and conditioned by several pro-inflammatory factors. Serotonin is one of these factors. It is secreted by the EC cells and it is the precursor of melatonin. Then EC cell proliferation leads to an increase in serotonin secretion. The balance between serotonin and melatonin depends on the expression of enzymes that regulate their synthesis and catabolism.

Furthermore, the synthesis and catabolism of melatonin, both in the colon and in the liver, can be modified by many factors, including pro-inflammatory cytokines.

Previous studies evidenced low levels (plasma, urine, salivary fluids) of melatonin in patients with functional and organic gastrointestinal pathologies compared to healthy controls [5,6] and they concluded that melatonin had a role in their pathogenesis. In the study by Radwan et al. [6] the concentration of 6-sulphatoxymelatonin (6-SMLT) was measured and statistically significant differences were found between patients with IBS (both with constipation and diarrhea) and healthy controls. For this reason, it has been already studied as an adjuvant treatment in various gastrointestinal pathologies such as irritable bowel syndrome (IBS), inflammatory bowel disease (IBD), and necrotizing enterocolitis [7].

However, melatonin synthesis probably plays a protective role for the colon mucosa due to its antioxidant, anti-inflammatory, and immunoregulatory activity, particularly in diseases based on the T lymphocyte response [30,31]. Previous studies show controversial results. Carpuso et al. [32] and Magro et al. [33] found a decrease in the concentration of serotonin in the colonic mucosa in patients with ulcerative colitis. However, a significant increase in both EC cells and colonic mucosal serotonin concentration was observed in experimental animals with ulcerative colitis [34,35]. Regardless of the factors contributing to the maintenance of the inflammatory process, the increased secretion of melatonin is a beneficial reaction. In previous studies, exogenous melatonin has also been found to have a positive effect on maintaining remission of ulcerative colitis [36], which is the main focus of pharmacotherapy in this disease. Earlier studies indicate that EC cell proliferation is present in the active phase of ulcerative colitis, regardless of the location of inflammatory changes in colonic mucosa [37]. In our study we also demonstrated that there is a negative correlation between the concentration of IR cells for serotonin and the extent of the disease, underlining the beneficial role of melatonin. In fact, melatonin secretion seems to be greater in patients with limited disease than in patients with ulcerative pancolitis. We also demonstrated a positive correlation between the levels of immunoreactive cells for serotonin and disease activity, and it should be another confirmation of the involvement of melatonin in mucosal inflammatory processes.

## 5. Conclusions

In our study, the decrease in the number of 5HT-iR cells in UC patients compared to healthy controls has been reported, and this reduction is significant in the rectum (*p* < 0.05).

Therefore, supplementation could be considered. In the case of severe disease, there is an increase in the expression of melatonin-secreting cells, probably as a compensatory response from the body to respond to the inflammation and oxidative stress.

The correlation analysis in UC patients shows that a higher expression of 5-HT-iR+ cells corresponds to a lower extension of the disease and a greater severity of the same.

Melatonin is difficult to quantify, probably because it is immediately released into the circulation and cannot be measured in the mucosa.

There is still no evidence that melatonin supplementation can be useful in the complex treatment of UC patients, as well as other digestive diseases, but our results provide a good background for future research in this field.

## Figures and Tables

**Figure 1 diagnostics-13-00204-f001:**
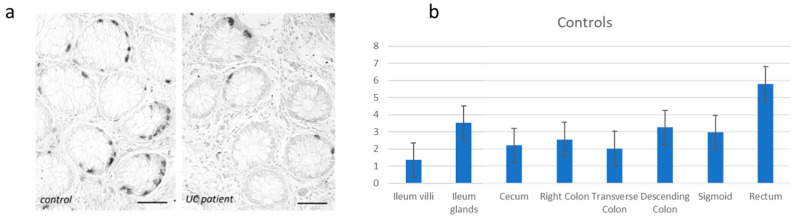
(**a**) Immunohistochemical evidence of 5-HT-iR cells in the healthy controls. Bar = 50 µm. (**b**) Distribution of the 5-HT-iR cells in the healthy controls.

**Figure 2 diagnostics-13-00204-f002:**
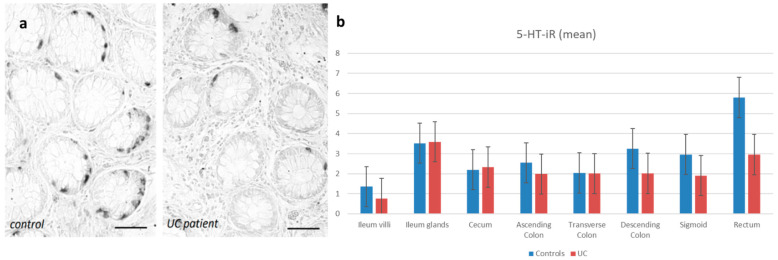
(**a**) Immunohistochemical evidence of 5-HT-iR cells in the rectum. Bar = 50 µ. (**b**) Distribution of the 5-HT-iR cells in the healthy controls and in the UC patients.

**Figure 3 diagnostics-13-00204-f003:**
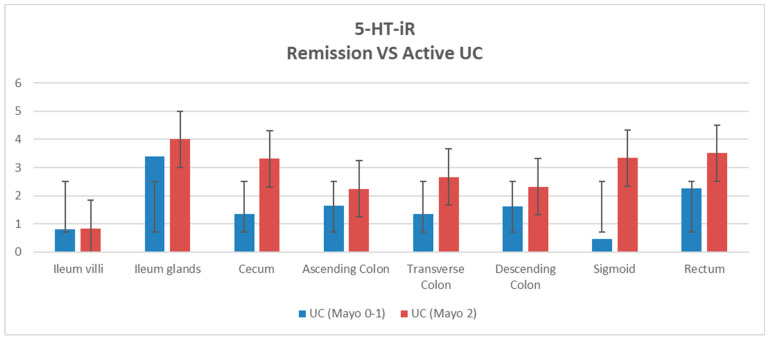
Distribution of the 5-HT-iR cells in the patients with UC in remission (Mayo 0–1) and active UC (Mayo 2).

**Figure 4 diagnostics-13-00204-f004:**
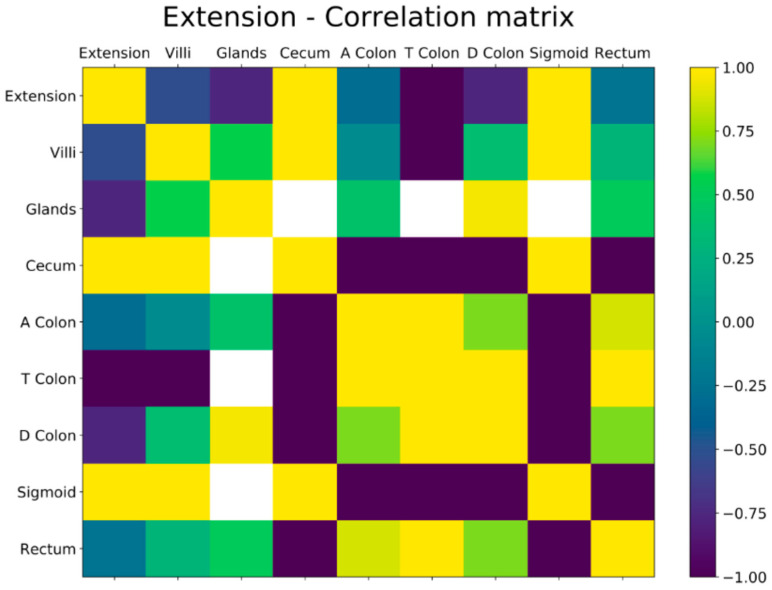
Correlation between the concentration of 5-HT-iR and the extension of the disease with Pearson’s coefficient.

**Figure 5 diagnostics-13-00204-f005:**
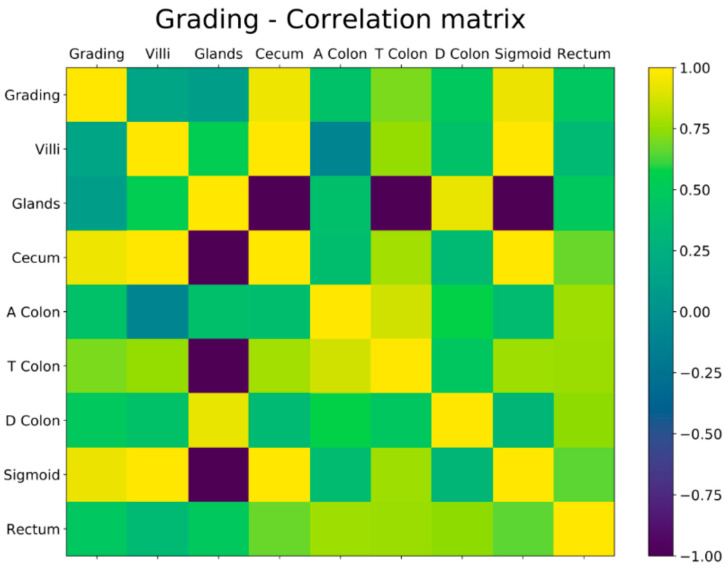
Correlation between the concentration of 5-HT-iR and the grading of the disease with Pearson’s coefficient.

**Table 1 diagnostics-13-00204-t001:** Demographics and endoscopic characteristics of the enrolled patients.

	N (%)	Sex (F)	Mean Age	Endoscopic Diagnosis (Mayo, *N*)	Extension (Montreal, *N*)
**Controls**	12 (57%)	3	58.8	-	-
**UC**	9 (43%)	3	48.7	Mayo 0 = 2	E1 = 2
				Mayo 1 = 2	E2 = 4
				Mayo 2 = 5	E3 = 1

## Data Availability

The datasets generated during and/or analyzed during the current study are available from the corresponding author on reasonable request.

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
