# Peer review of "Serotonin and Melatonin in Human Lower Gastrointestinal Tract"

_diagnostics, 2023, doi:10.3390/diagnostics13020204_

Round 1

Reviewer 1 Report

The analysed article regards a very interesting and specific theme of research, very well documented and with many perspectives as future applications in clinics. 

In order to improve the perception, readability and scientific soundness of this study, I do  consider that the authors must add a series of important data and should rethink some of the graphics displayed.

No information or insufficient data is given regarding the study development, the particular hospital that housed the entire, where was the study conducted, the patients' provenience and accurate data regarding their pathology, who performed the evaluation of the patients and acknowledged their pathological status, we get no details about the laboratory procedures, kits and reagents manufacturers and so many details that could make the performed evaluations reproducible.

The graphics are not concludent since the OY presents only numbers, without being explained what they designate. 

The discussions should contain parallels with different clinical studies that performed similar experimental procedures and also, it should be mentioned the outcome of clinical studies that included patients supplemented with melatonin, for particular GIT disorders.

I do consider that the authors can and should increase the quality of this interesting article and the innovative idea behind the design. 

Author Response

No information or insufficient data is given regarding the study development, the particular hospital that housed the entire, where was the study conducted, the patients' provenience and accurate data regarding their pathology, who performed the evaluation of the patients and acknowledged their pathological status, we get no details about the laboratory procedures, kits and reagents manufacturers and so many details that could make the performed evaluations reproducible.

Thank you very much for this comment. We addedd these informations in the material and methods section.

The graphics are not concludent since the OY presents only numbers, without being explained what they designate. 

Thanks for this suggestion. We improved the graphics and we added the high resolution images in the supplementary material.

The discussions should contain parallels with different clinical studies that performed similar experimental procedures and also, it should be mentioned the outcome of clinical studies that included patients supplemented with melatonin, for particular GIT disorders.

Thank you very much for this suggestion. We added these informations in the discussion.

I do consider that the authors can and should increase the quality of this interesting article and the innovative idea behind the design.

Reviewer 2 Report

The title of this article is “Serotonin and melatonin in human lower gastrointestinal tract”. This is an interesting topic, however, there are still some areas of the article that need to be revised:

1. The "Material and methods" section of the article. The author needs to further organize this section, such as adding some new subheadings or merging some similar content to make the article more concise.

2. The layout of the images, tables and icons in the article needed to be adjusted. A confusing layout makes the article look unattractive and unreadable.

3. Figure 4. Correlation between the concentration of 5-HT-iR and the extension of the disease with Pearson’s coefficient. In this section, the authors need to explore the results in more depth and check the results obtained with recently published journals.

4. Discussion section of the article. The authors conclude that alterations in melatonin in the colon have some association with inflammatory processes, and for this conclusion, the authors need to parse it in the context of the mechanisms of melatonin's effect on the alleviation of inflammation.

5. Please revise the English expressions in your essay by removing unnecessary "the" from the sentences, making sure the sentences look more concise, and replacing words that appear too often in the text.

6. Authors are requested to carefully check the format of the references used in the article to ensure that the references are in the required format.

Author Response

The title of this article is “Serotonin and melatonin in human lower gastrointestinal tract”. This is an interesting topic, however, there are still some areas of the article that need to be revised:

  1. The "Material and methods" section of the article. The author needs to further organize this section, such as adding some new subheadings or merging some similar content to make the article more concise.

Thank you very much for your comment. We reorganized the material and methods section.

  1. The layout of the images, tables and icons in the article needed to be adjusted. A confusing layout makes the article look unattractive and unreadable.

Thanks for this suggestion. We improved the graphics and we added the high resolution images in the supplementary material.

  1. Figure 4. Correlation between the concentration of 5-HT-iR and the extension of the disease with Pearson’s coefficient. In this section, the authors need to explore the results in more depth and check the results obtained with recently published journals.

Thank you for this valuable suggestion. We compared the results in the discussion.

  1. Discussion section of the article. The authors conclude that alterations in melatonin in the colon have some association with inflammatory processes, and for this conclusion, the authors need to parse it in the context of the mechanisms of melatonin's effect on the alleviation of inflammation.

Thank you. We added informations on the melatonin’s role on the alleviation of inflammation.

  1. Please revise the English expressions in your essay by removing unnecessary "the" from the sentences, making sure the sentences look more concise, and replacing words that appear too often in the text.

Thank you for your comment. We revised the english expressions.

  1. Authors are requested to carefully check the format of the references used in the article to ensure that the references are in the required format.

Thank you. We checked the references.

Round 2

Reviewer 1 Report

Dear authors,

Figures 1, 2, 3 are still misleading, no units of measure, and no details are presented in order to understand the meaning of the numbers. The comma in English should be replaced with a point, in order to express a fraction of a unit.

It is not clear if the clinical procedures took place in a hospital or the UNIVERSITY of ROME? Please correct the inadvertences, if any.

Author Response

Dear reviewer thanks for your respectable suggestion. We changed the graphics. As we exlplained in the paragraph "cell counts" the numbers represent percentage of cells. More in detail, a count of total enterocytes and serotonin immunoreactive endocrine cells was performed. The ratio of 5HT-ir endocrine cells / total enterocytes was obtained for each field and an average was made of the values ​​obtained in the fields.

The hospital in which the experiments were performed is a university hospital.

Round 3

Reviewer 1 Report

The article is accepted for publication in the present form.